# Sex and circadian regulation of metabolic demands in the rat kidney: A modeling analysis

Pritha Dutta[1]*, Anita T. Layton[1,2,3,4]

**1** Department of Applied Mathematics, University of Waterloo, Waterloo, Ontario, Canada, **2** Cheriton School of Computer Science, University of Waterloo, Waterloo, Ontario, Canada, **3** Department of Biology, University of Waterloo, Waterloo, Ontario, Canada, **4** School of Pharmacy, University of Waterloo, Waterloo, Ontario, Canada

* p7dutta@uwaterloo.ca

**Data Availability Statement:** The code and data used for this study can be accessed at https://github.com/Pritha17/sex-and-circadian-nephron-model.

## Abstract

Renal hemodynamics, renal transporter expression levels, and urine excretion exhibit circadian variations. Disruption of these diurnal patterns is associated with the pathophysiology of hypertension and chronic kidney disease. Renal hemodynamics determines oxygen delivery, whereas renal transport and metabolism determines oxygen consumption; the balance between them yields renal oxygenation which also demonstrates 24-h periodicity. Another notable modulator of kidney function is sex, which has impacts on renal hemodynamics and transport function that are regulated by as well as independent of the circadian clock. The goal of this study was to investigate the diurnal and sexual variations in renal oxygen consumption and oxygenation. For this purpose, we developed computational models of rat kidney function that represent sexual dimorphism and circadian variation in renal hemodynamics and transporter activities. Model simulations predicted substantial differences in tubular Na$^+$ transport and oxygen consumption among different nephron segments. We also simulated the effect of loop diuretics, which are used in the treatment of renal hypoxia, on medullary oxygen tension. Our model predicted a significantly higher effect of loop diuretics on medullary oxygenation in female rats compared to male rats and when administered during the active phase.

## Introduction

The kidneys maintain chemical and fluid homeostasis in the body by excreting waste products and excess solutes and fluids. As a result, the kidneys receive a high blood flow, approximately 25% of the cardiac output. In addition, the kidneys consume the second highest amount of oxygen, normalized by organ weight, after the heart. However, compared to other organs, renal oxygen extraction is very low, approximately 10–15%, whereas that for the heart is about 45%. This makes the kidneys susceptible to hypoxia, which plays an important role in the development of acute or chronic kidney diseases [1–5].

**Funding:** This study is supported in part by grants from the Natural Sciences and Engineering Research Council (NSERC) and Canadian Institutes of Health Research (CIHR) of Canada to A.T. Layton. The funders had no role in study design, data collection and analysis, decision to publish, or preparation of the manuscript.

**Competing interests:** The authors have declared that no competing interests exist.

Renal blood flow is highly heterogeneous, as the cortex receives a sufficient amount while only 10–15% of the perfusion goes to the medulla. This low perfusion is required to maintain osmotic gradients and urine concentrating ability of the medulla [6]. However, medullary oxygen consumption accounts for approximately 20% of the total oxygen consumption in the kidney. The medullary thick ascending limbs have a high demand for oxygen to conduct $Na^+$ transport but low oxygen delivery, making this segment particularly prone to hypoxia [6–8]. The oxygen shunting between the descending and ascending vasa recta in the medulla also contributes to the low oxygen availability in this region [6–8]. Additionally, tubular cells within the S3 segment of the proximal tubule, which is in the outer medullary region, have high oxygen demand due to the abundance of active $Na^+$-$K^+$-ATPases. In addition, an increased demand for oxygen in the kidney cannot be met by an increased blood flow, as this would also increase the GFR and tubular $Na^+$ load. Since $Na^+$ reabsorption is the major renal oxygen consuming process, increased renal blood flow also increases the oxygen demand. Thus, increased oxygen delivery is counteracted by increased oxygen consumption [9]. The combination of these factors can aggravate outer medullary hypoxia even in healthy kidneys [2, 6, 8, 10].

In recent years, two new dimensions have emerged for investigation of kidney function: sex and time-of-day. Sex hormones regulate the structure and function of the kidneys [11, 12]. Studies have shown that the expression of membrane transporters also differ between male and female in rodent kidneys [13]. For instance, female rats express lower $Na^+$/$H^+$ exchanger 3 (NHE3) in the proximal tubules compared to the male counterparts and thus reabsorb a substantially lower fraction of the filtered $Na^+$. The distal tubular segments in female rats handle the resulting higher fractional $Na^+$ delivery by augmenting the abundance and activity of transporters, such as $Na^+$-$K^+$-$Cl^-$ cotransporter 2 (NKCC2) and $Na^+$-$Cl^-$ cotransporter (NCC) in the thick ascending limbs and distal convoluted tubules. Female rats exhibit higher NKCC2 and NCC activity (almost double) relative to males. Male rats transport a larger fraction of the filtered $Na^+$ through NHE3 in the proximal tubules, whereas female rats transport a larger fraction of the filtered $Na^+$ through NKCC2 in the medullary thick ascending limbs [13–16]. Thus, cortical and medullary oxygen consumptions are different between the sexes. In addition, different segments have different transport efficiency- a segment having higher paracellular transport would transport more $Na^+$ moles per mole of $O_2$ consumed [17, 18]. Hence, a goal of this study is to investigate the sex differences in whole-kidney and regional oxygen consumption and segmental transport efficiency, which we hypothesize would arise from the difference in segmental distribution of $Na^+$ transport between males and females.

Renal transporter proteins, including NHE3, NKCC2, NCC, and epithelial $Na^+$ channels (ENaC) are regulated by clock proteins, brain and muscle ARNT-like 1 (BMAL1) and period circadian regulator 1 (PER1) [19–22], which cause significant diurnal rhythms in GFR, filtered solute loads, and urinary solute and volume excretion [23–25]. Sex and time-of-day should not be considered as two independent regulators of kidney function. For instance, notable sex differences have been reported in the regulation of $Na^+$ transport by BMAL1 [26]. Thus, any investigation of kidney metabolism and function must take into account sex and time-of-day as variables.

Several models have been developed that capture sex differences in renal hemodynamics and transporter expressions in rats [14, 15], mice [27, 28], and humans [29, 30]. Among these models, only one has considered diurnal variations in transporter activities and kidney function [27]. In addition, none of these models have looked at the circadian variation in oxygen consumption and oxygenation between the sexes.

To assess the sex- and time-of-day specific differences in oxygen consumption along different nephron segments, we developed computational models of circadian regulation of solute and water epithelial transport in male and female rats (Fig 1). We have previously developed

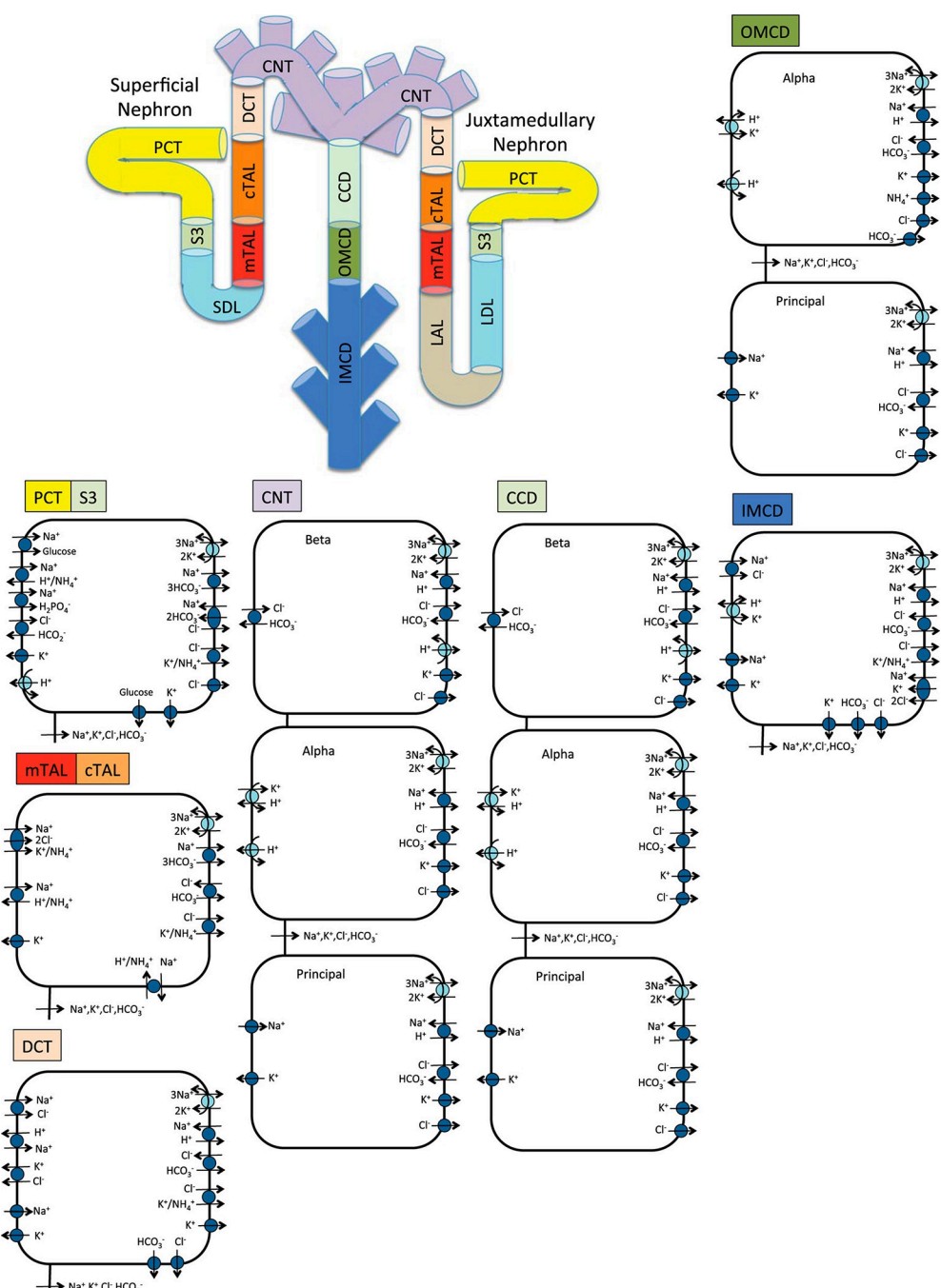

**Fig 1. Schematic diagram of the nephron system (not to scale).** The diagram shows the superficial nephron and one juxtamedullary nephron (the model includes five representative juxtamedullary nephrons). The model considers the transport of water and 15 solutes along each nephron. The diagram displays only the main $Na^+$, $K^+$, and $Cl^-$ transporters. PCT, proximal convoluted tubule; SDL, short descending limb; mTAL, medullary thick ascending limb; cTAL, cortical thick ascending limb; DCT, distal convoluted tubule; CNT, connecting tubule; CCD, cortical collecting duct; OMCD, outer-medullary collecting duct; IMCD, inner-medullary collecting duct; LDL/LAL, thin descending/ascending limb. Adopted from [35].

epithelial transport models of the rat kidney that are sex specific [14–16, 31, 32] or that incorporate circadian rhythms in GFR and transport activities [33, 34], but none that considers both variables. Using the sex- and time-of-day specific models, we simulated the circadian and sexual variation in $Na^+$ transport and oxygen variation along different nephron segments and regions (cortical and medullary regions). The model predicted significant differences in $Na^+$ transport and oxygen consumption along different nephron segments and between male and female rats. We also developed an equation to compute renal oxygen tension based on the model's predicted oxygen consumption. Using this equation, we predicted the change in outer medullary oxygenation on treatment with loop diuretics, which are commonly used to treat renal hypoxia. Model simulations demonstrated a significantly higher effect of loop diuretics on renal oxygenation in female rats compared to male rats and when administered during the active phase.

## Results

We conducted model simulations to predict solute and volume transport along nephron segments in male and female rats. GFR and filtered $Na^+$ and $K^+$ loads, which were used as inputs to predict fluid and solute flows along different nephron segments, are shown in S1 Fig. The circadian regulated GFR and transporter activities are shown in Fig 2 (values normalized to mean). The predicted $Na^+$, $K^+$, $Cl^-$, and volume deliveries to different nephron segments (proximl tubules, thick ascending limbs, distal convoluted tubules, connecting tubules, and collecting ducts) at different zeitgeber times in male and female rats are shown in Fig 3. The solute and volume deliveries show a diurnal rhythm, peaking in the dark (active) phase, and is in phase with the GFR. The rat model predicted that more than half of the filtered $Na^+$, $K^+$, $Cl^-$, and volume are reabsorbed along the proximal tubules with the thick ascending limbs reabsorbing most of the remaining solutes and volume. $K^+$ deliveries to the connecting tubules and collecting ducts increase because the transepithelial elctrochemical gradient causd by $Na^+$ reabsorption through ENaC in the distal convoluted tubules and connecting tubules favours $K^+$ secretion.

### $Na^+$ transport exhibits significant sex-, time-of-day, and regional variations

The predicted $T_{Na}$ (active, passive, and total) in the proximal tubules, thick ascending limbs, and distal tubules (comprising distal convoluted tubules, connecting tubules, and collecting ducts) of male and female rats at different zeitgeber times (2, 6, 14, and 18 h) are shown in Fig 4. The predicted active, passive, and total $T_{Na}$ display diurnal rhythms in phase with GFR, peaking during the active phase (ZT14). Both transcellular and paracellular $T_{Na}$ change proportionally with luminal flow. Total $T_{Na}$ increases by 47%, 34%, and 16% during the active phase relative to the inactive phase in the male proximal tubules, thick ascending limbs, and distal tubules, respectively. The corresponding total $T_{Na}$ increments in the female nephron segments are 38%, 49%, and 26%, respectively. Proximal tubule active $T_{Na}$ is higher in males compared to females because males have higher NHE3 activity and filtered $Na^+$ in the proximal tubules. In addition, proximal tubule passive $T_{Na}$ is higher in males than in females. This combined higher active and passive $T_{Na}$ results in lower $Na^+$ delivery to the male thick ascending limbs relative to females. Thus, female thick ascending limbs have higher active $T_{Na}$ due to higher $Na^+$ delivery as well as higher NKCC2 and $Na^+$-$K^+$-ATPase activities.

Given that the renal medulla is poorly perfused compared to the cortex, we analyzed $T_{Na}$ for the two regions separately. Fig 5 shows the predicted $T_{Na}$ for the cortical segments, medullary segments, and whole kidney. The predicted total $T_{Na}$ in the cortical segments (comprising the proximal convoluted tubules, cortical thick ascending limbs, distal convoluted tubules,

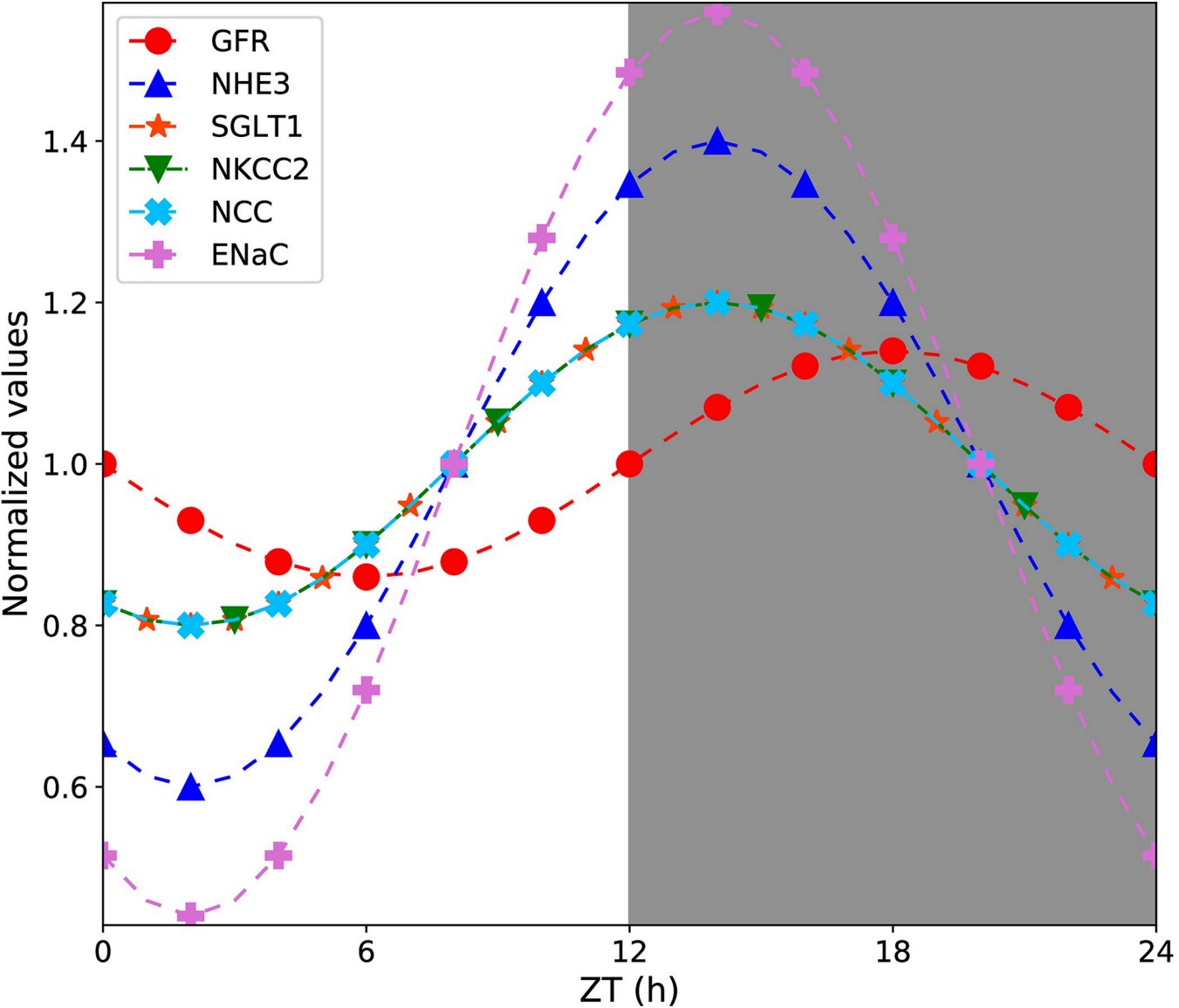

**Fig 2. Diurnal variation in GFR and transporter activities regulated by circadian clocks.** Diurnal oscillations in glomerular filtration rate (GFR), Na$^+$/H$^+$ exchanger 3 (NHE3), sodium-glucose transporter 1 (SGLT1), Na$^+$-K$^+$-Cl$^-$ cotransporter 2 (NKCC2) activity, Na$^+$-Cl$^-$ cotransporter (NCC), and epithelial sodium channel (ENaC) activities. Values are normalized to mean values.

connecting tubules, and cortical collecting ducts) is ~51% higher in the male rats compared to female rats during both active and inactive phases. This is because in the cortical region, the majority of Na$^+$ transport occurs in the proximal convoluted tubule, and male rats have higher Na$^+$ transport in the proximal tubules. The predicted total T$_{Na}$ in the medullary segments (comprising proximal straight tubule, medullary thick ascending limbs, and outer and inner medullary collecting ducts) is ~16% higher in female rats compared to male rats during both active and inactive phases. This is because in the medullary region, the majority of Na$^+$ transport occurs in the medullary thick ascending limbs, and female rats have higher Na$^+$ transport in the thick ascending limbs.

Paracellular transport, which is passive diffusion driven by transepithelial electrochemical gradient, is the mechanism for efficient oxygen utilization as it does not require energy from

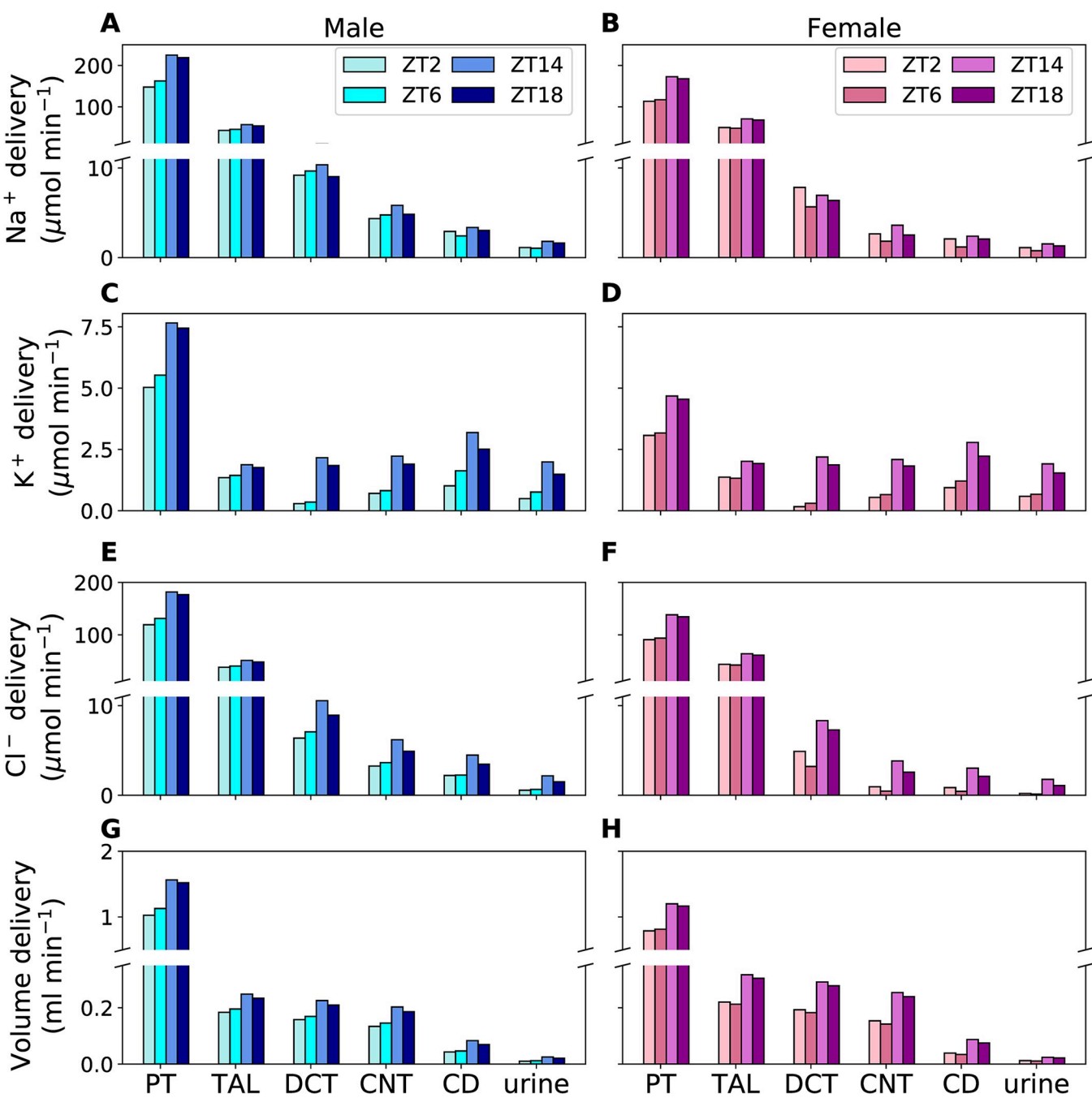

**Fig 3. Predicted solute and volume deliveries and urinary excretions.** Predicted $Na^+$ (A, B), $K^+$ (C, D), $Cl^-$ (E, F), and volume (G, H) deliveries to the proximal tubules (PT), thick ascending limbs (TAL), distal convoluted tubules (DCT), connecting tubules (CNT), and collecting ducts (CD) and urinary excretions of male and female rats at zeitgeber times 2, 6, 14, and 18 h. The values are given per kidney.

ATP hydrolysis. Paracellular $Na^+$ transport (passive $T_{Na}$) in the proximal tubule follows a diurnal pattern in phase with the GFR and it is higher in the male rats due to higher $Na^+$ delivery. Paracellular transport is almost zero or results in $Na^+$ secretion in the thick ascending limbs due to the following reason. In the initial part of the medullary thick ascending limb, both luminal and interstitial $Na^+$ concentrations are high and the lumen-positive potential is

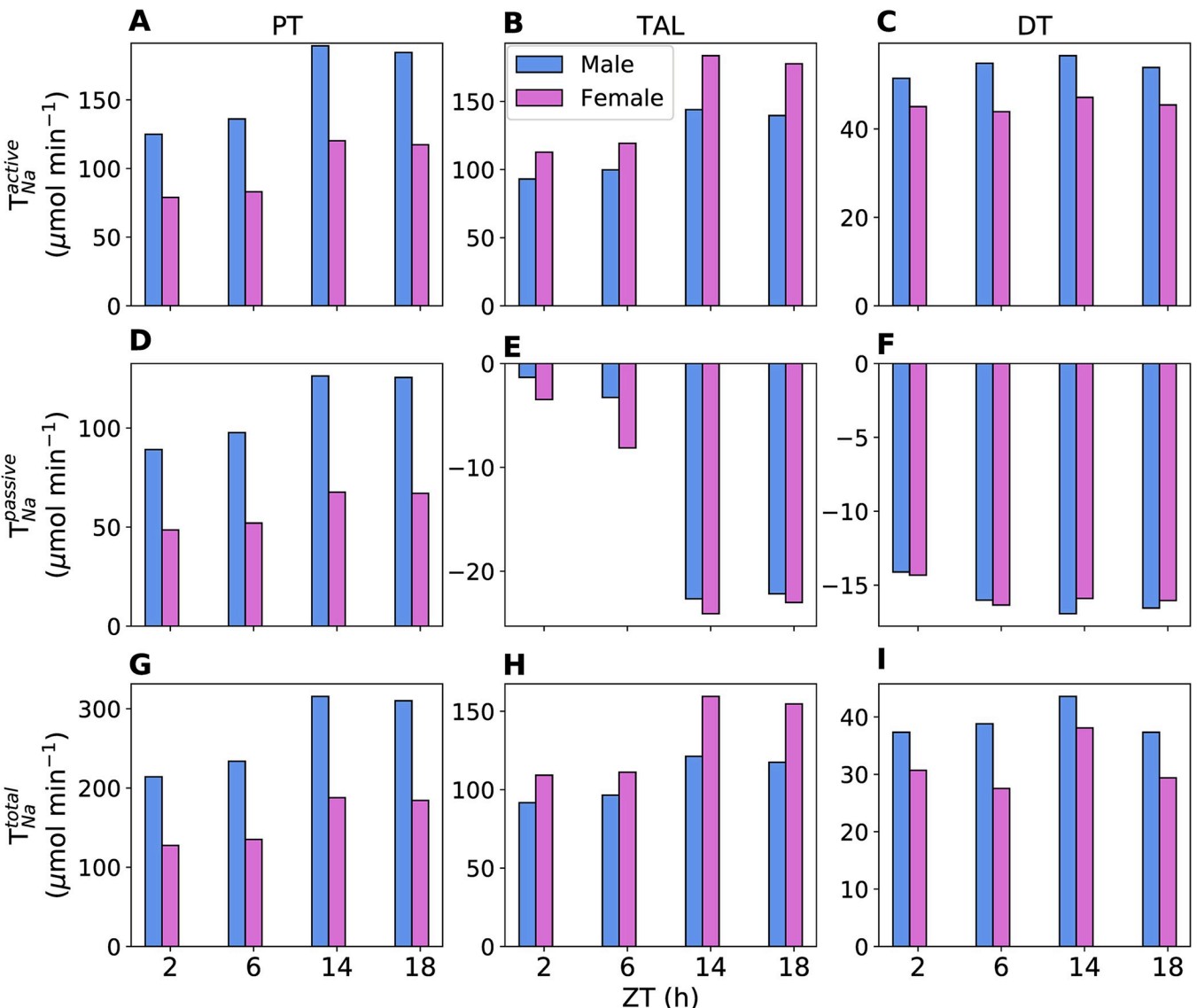

**Fig 4. Predicted segmental renal $T_{Na}$.** Predicted renal active (A, B, C), passive (D, E, F), and total (G, H, I) $T_{Na}$ in the proximal tubules (PT), thick ascending limbs (TAL), and distal tubules (DT) of male and female rats at zeitgeber times 2, 6, 14 and 18 h. The values are given per kidney.

responsible for transporting $Na^+$ from the lumen to the interstitium. However, as $Na^+$ is reabsorbed, the concentration gradient for $Na^+$ increases which causes $Na^+$ to enter the lumen from the interstitium [36].

## $Q_{O2}$ exhibits significant sex-, time-of-day, and regional variations

The predicted total $Q_{O2}$ in the proximal tubules, thick ascending limbs, and distal tubules of male and female rats at different zeitgeber times (2, 6, 14, and 18 h) are shown in Fig 6A–6C. Fig 6D–6F shows these predicted values for the cortical segments, medullary segments, and whole kidney. Since, our model assumes passive $Q_{O2}$ to be constant, total $Q_{O2}$ changes proportionally with active $T_{Na}$. Male rats have higher oxygen consumption in the proximal tubules because of higher luminal flow and NHE3 activity, whereas female rats have higher oxygen consumption in the thick ascending limbs due to higher NKCC2 and $Na^+$-$K^+$-ATPase activity.

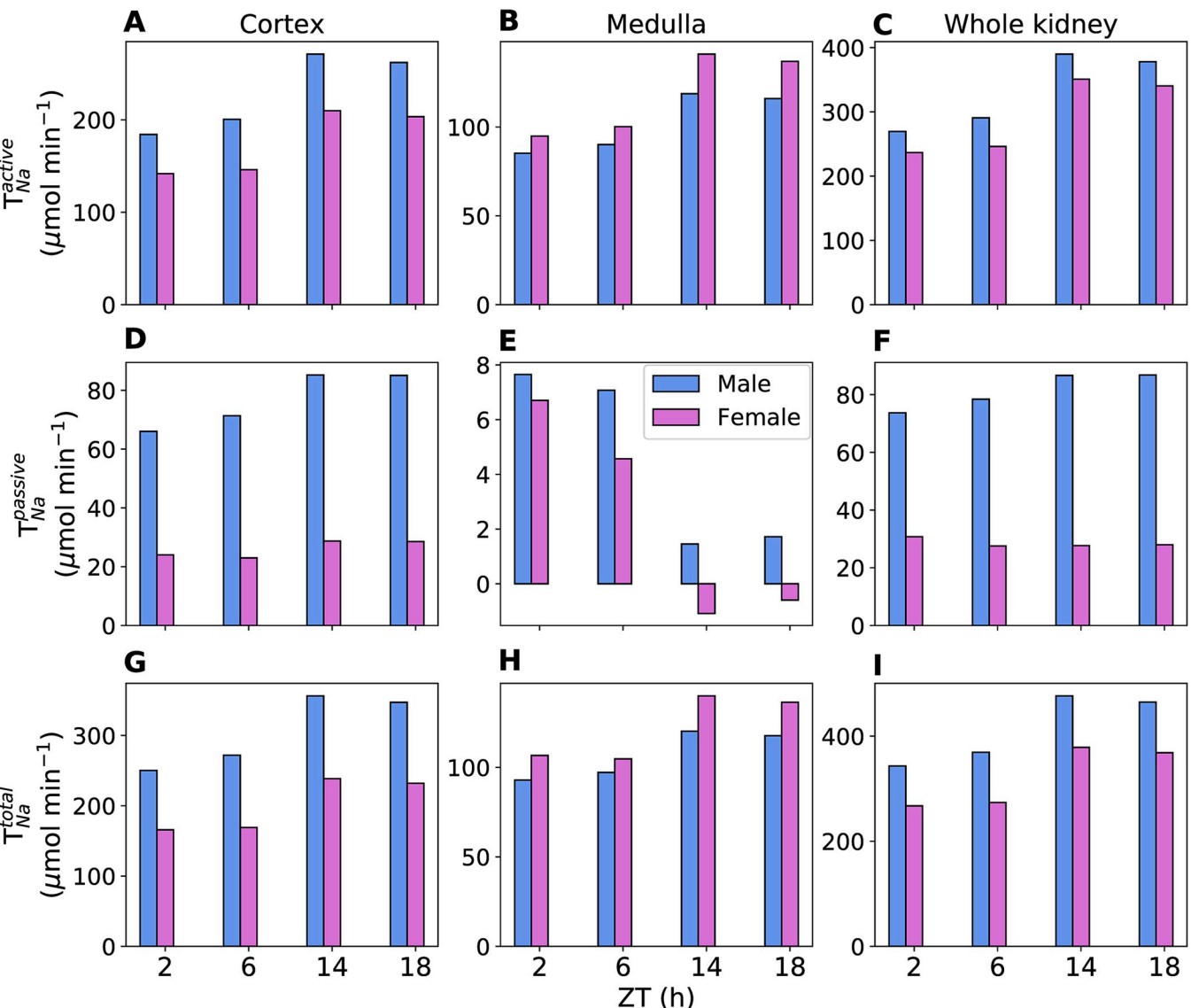

**Fig 5. Predicted regional renal $T_{Na}$.** Predicted renal active (A, B, C), passive (D, E, F), and total (G, H, I) $T_{Na}$ in the cortical segments, medullary segments, and whole kidney of male and female rats at zeitgeber times 2, 6, 14 and 18 h. The values are given per kidney.

Similarly, oxygen consumption in the renal cortex is ~1/3 higher in male rats and oxygen consumption in the renal medulla is ~1/5 higher in female rats during both the active and inactive phases. With respect to the whole kidney, male rats have ~12% higher oxygen consumption than female rats. Since male rat kidneys filter ~25% more $Na^+$ than the female counterparts, their whole kidney $Q_{O2}$ is expected to be ~25% higher than that of female rats. However, male rats transport more $Na^+$ along the proximal tubule, whereas female rats transport more $Na^+$ along the thick ascending limbs. Proximal tubules have higher $Na^+$ transport efficiency compared to the other nephron segments (S2 Fig) due to their high paracellular $Na^+$ transport. This could be a possible reason for the lower whole kidney $Q_{O2}$ difference between male and female rats.

The efficiency of oxygen utilization for $Na^+$ reabsorption varies between tubular segments. It is higher in the proximal tubules than in the thick ascending limbs and distal tubules (S2 Fig). This is because the net paracellular transport, which does not require ATP hydrolysis and

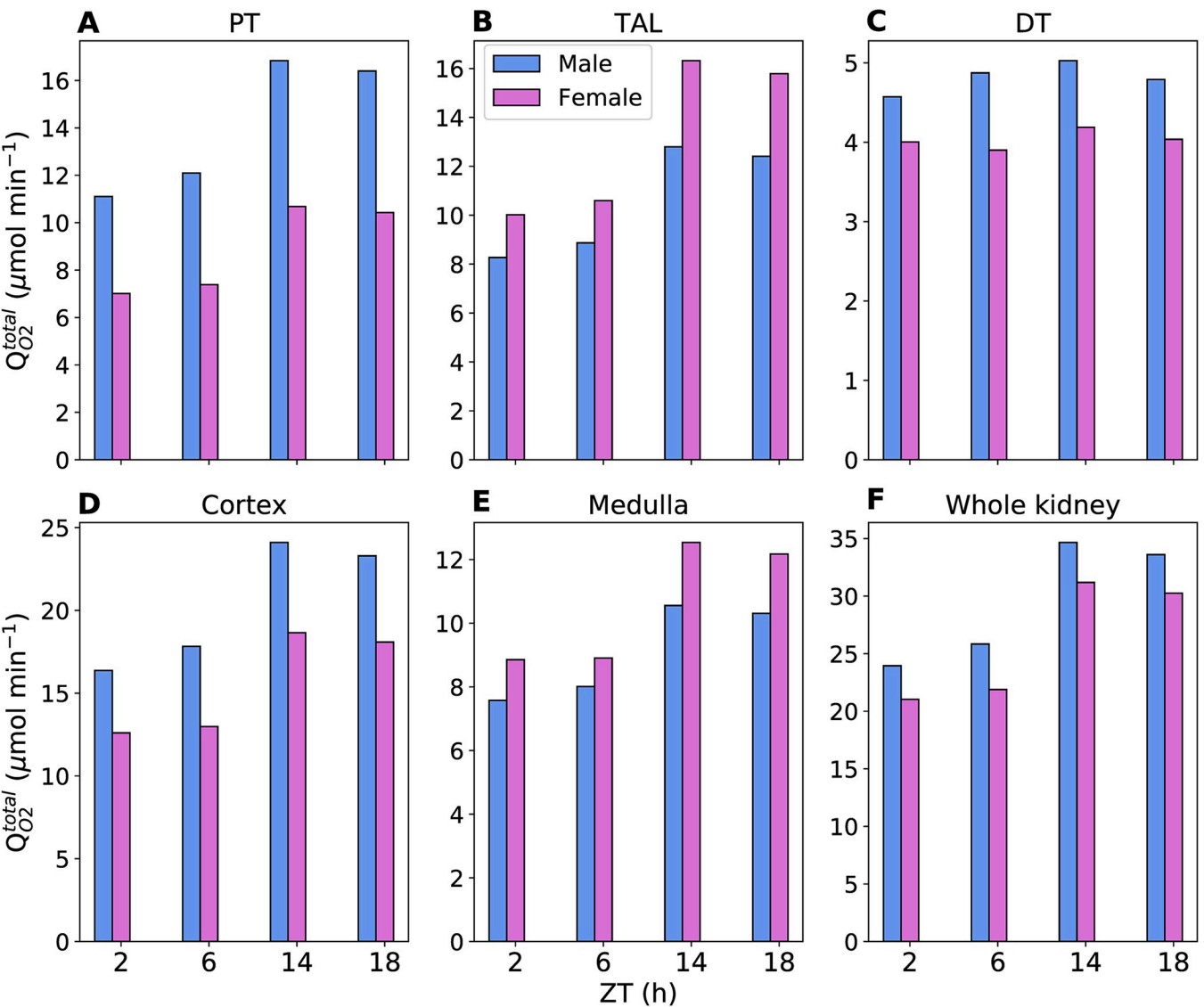

**Fig 6. Predicted segmental and regional renal $Q_{O2}$.** Predicted renal total $Q_{O2}$ in (A, B, C) the proximal tubules (PT), thick ascending limbs (TAL), and distal tubules (DT); and (D, E, F) the cortical segments, medullary segments, and whole kidney of male and female rats at zeitgeber times 2, 6, 14 and 18 h. The values are given per kidney.

hence is an important determinant of oxygen utilization efficiency, is almost zero in the thick ascending limbs and distal tubules. Transcellular $T_{Na}$ changes proportionally with luminal flow. Thus, as GFR increases, $T_{Na}^{active}$ (and thus $Q_{O2}^{active}$) increases at the same rate as $T_{Na}^{total}$. Thus, the model predicts that the oxygen utilization efficiency does not change significantly during the day (S2 Fig).

## Loop diuretics have significantly greater effect on medullary oxygenation in female rats

Based on the predicted $Q_{O2}$, we computed $p_{O2}$ using Eq 4. The predicted and experimental $p_{O2}$ (discussed in Methods) values are given in Fig 7. The experimental outer medullary $p_{O2}$ was calculated using the peak time and oscillation amplitude of 13 h (ZT) and 8%, respectively [37].

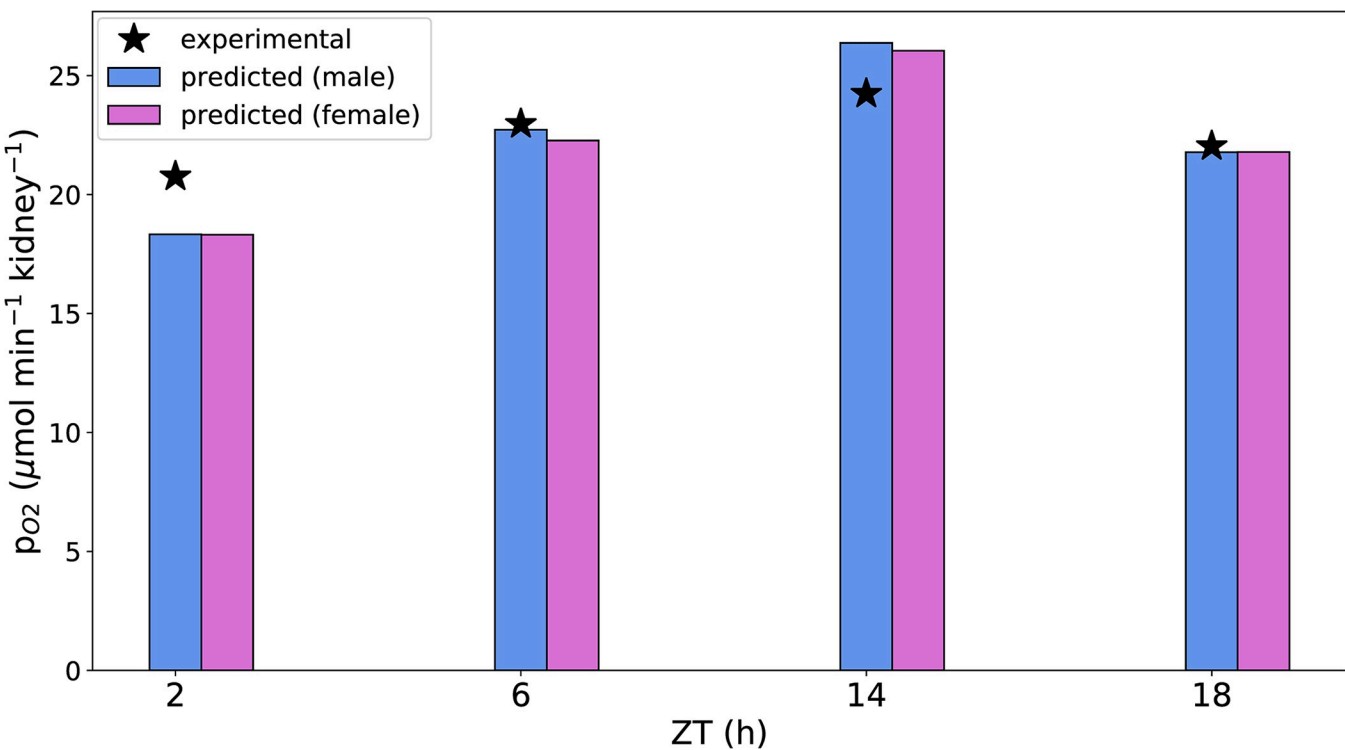

**Fig 7. Comparison of experimental and predicted $p_{O_2}$.** Experimental and predicted $p_{O_2}$ in the outer medullary region of male and female rats at zeitgeber times 2, 6, 14, and 18 h.

The model predicted an ~43% increase in $p_{O_2}$ during the active phase (ZT14) relative to the inactive phase (ZT2) for both male and female rats; the corresponding increase in the experimental $p_{O_2}$ data is ~16%. This discrepancy is possibly due to our $D_{O_2}$ calculation (Eq 5). Due to the unavailability of data on diurnal variation in $D_{O_2}$, we estimated it based on the diurnal variation in renal blood flow. Renal blood flow is ~40% higher during the active phase compared to the inactive phase [38]. This contributes to the high diurnal variation in our predicted $p_{O_2}$.

We then used the rat models to simulate the effect of loop diuretics on oxygen consumption and renal oxygen tension. By inhibiting thick ascending limb active $Na^+$ transport, loop diuretics have been found to ameliorate medullary hypoxia [39]. Fig 8A shows the medullary oxygen consumption before and after intervention with loop diuretics during the inactive (ZT2) and active (ZT14) phases. Loop diuretics induce similar fractional reductions in oxygen consumption in both sexes: by ~9.2% during the inactive phase and by ~8.4% during the active phase in both sexes.

A reduction in $Q_{O_2}$ is followed by an increase in $p_{O_2}$, which exhibits much more significant sex difference. For male rats, $p_{O_2}$ increased by 4.1% and 7.8% during the inactive and active phases, respectively, whereas, for female rats, the corresponding increases were 5.4% and 21%, respectively. Loop diuretics inhibit NKCC2 activity, which in turn lowers $Na^+$-$K^+$-ATPase activity. Since female rats have higher NKCC2 and $Na^+$-$K^+$-ATPase activity (two times higher) than male rats, they have higher increase in medullary oxygen tension with loop diuretics. Thus, loop diuretics have significantly greater effect on oxygenation in female rats than in male rats.

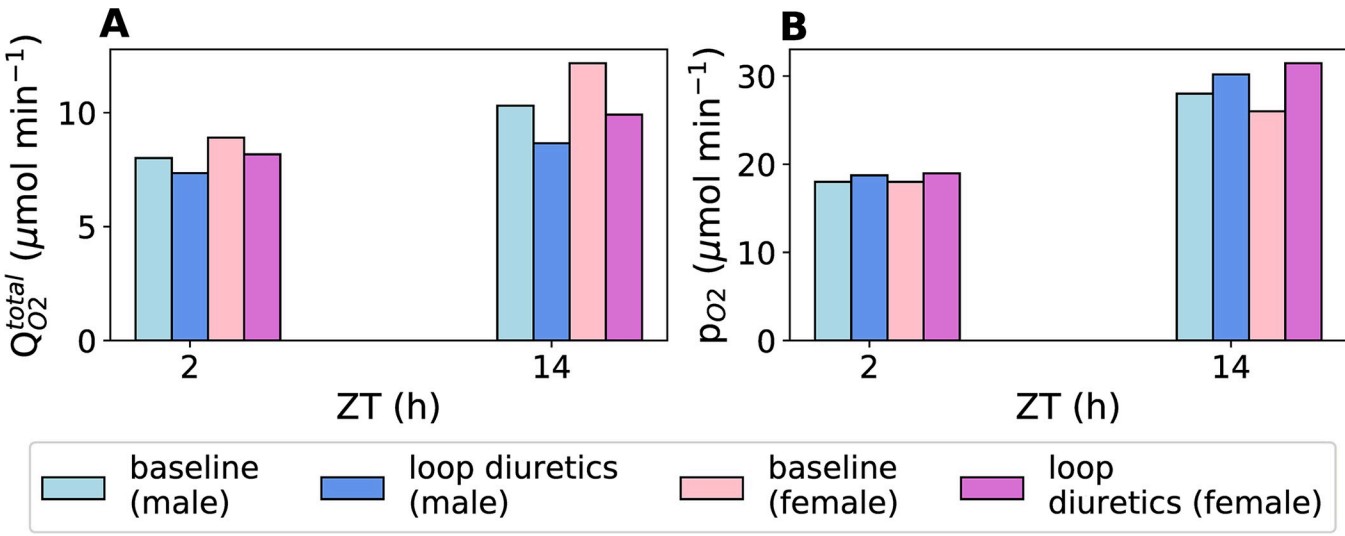

**Fig 8. Predicted medullary $Q_{O2}$ and $p_{O2}$ before and after inhibition with loop diuretics.** (A) Predicted total $Q_{O2}$ before and after inhibition with loop diuretics in the medullary segments of male and female rats at zeitgeber times 2 h (inactive phase) and 14 h (active phase). (B) Predicted renal medullary oxygen tension before and after inhibition by loop diuretics in male and female rats at zeitgeber times 2 h (inactive phase) and 14 h (active phase).

## ENaC inhibition has greater diuretic and natriuretic responses in male rats

Soliman et al. reported that ENaC inhibition caused greater diuretic and natriuretic responses in male rats compared to female rats at the beginning of both active and inactive phases [40]. We simulated the effect of ENaC inhibition by inhibiting ENaC activity in our model by 100%. Fig 9 compares the predicted fractional increases in volume and $Na^+$ excretions after ENaC inhibition at the beginning of the inactive (ZT = 0 h) and active (ZT = 12 h) phases in male and female rats with the corresponding fractional increases reported by Soliman et al. [40]. ENaC inhibition causes higher fractional increases in volume and $Na^+$ excretions in male rats compared to female rats during the beginning of both inactive and active phases. This is because males have higher delivered volume and $Na^+$ loads to the connecting tubule, the segment from which ENaC is the main $Na^+$ transporter. The $Na^+$ loads delivered to the male connecting tubule are 98% and 61% higher during zeitgeber times 0 and 12 h, respectively, compared to that delivered to the female connecting tubule. The corresponding volume loads delivered to the male connecting tubule are 85% and 57% higher, respectively. For this reason, inhibiting ENaC causes a greater diuretic and natriuretic response in male rats compared to females. Another interesting observation is that the diuretic and natriuretic responses are significantly higher when ENaC is inhibited at the beginning of the inactive phase compared to the beginning of the active phase in male rats. This is because the $Na^+$ and volume loads delivered to the connecting tubule are 35% and 28% higher, respectively, during ZT = 0 h compared to ZT = 12 h. By contrast, for females, the corresponding loads are only 9% and 4.6% higher, respectively. This is because the GFR is the same at ZT = 0 h and ZT = 12 h (Fig 2). However, the $Na^+$ transporters (NHE3, NKCC2, NCC) are near nadir at 0 h (Fig 2). Hence, less $Na^+$ and water gets reabsorbed along the early nephron segments at 0 h. Since, males have 25% higher GFR than females, the delivered load to the connecting tubule is significantly higher in male rats.

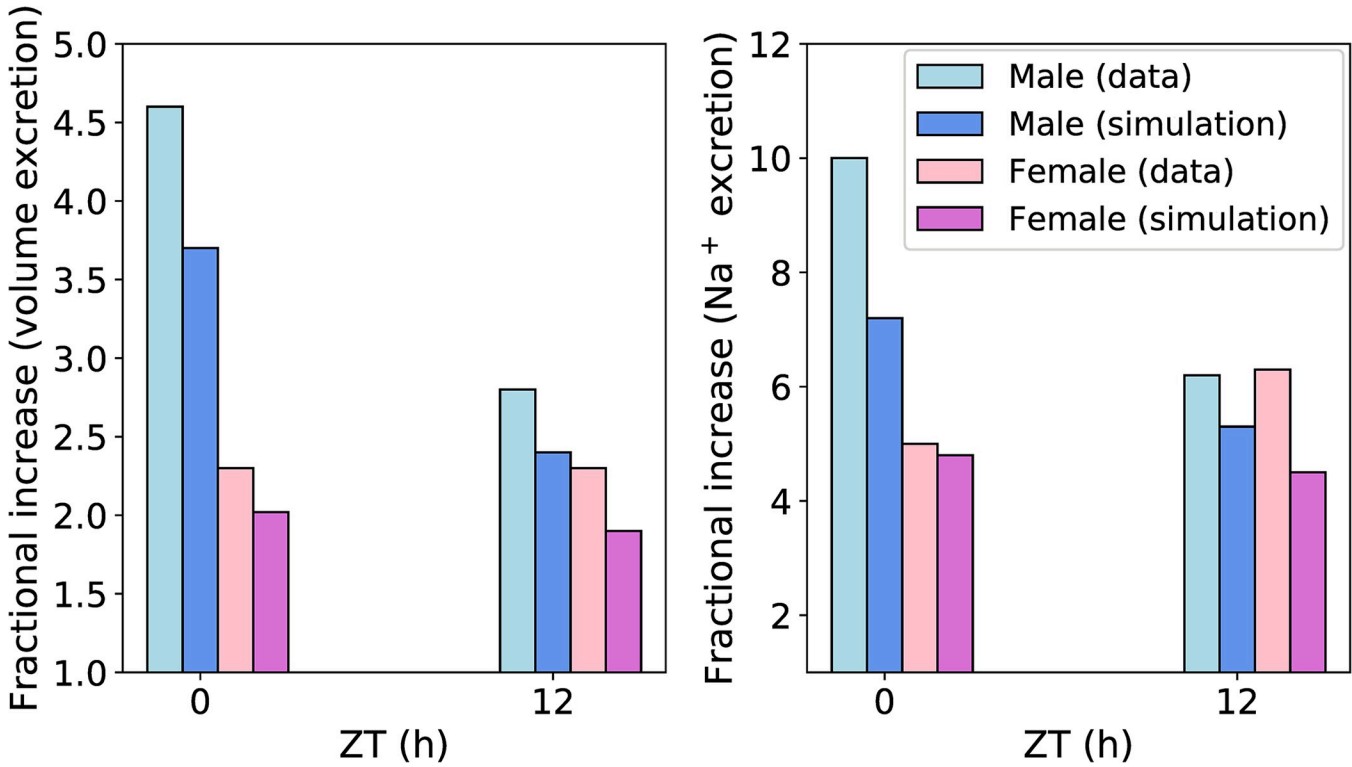

**Fig 9. Comparison of experimental and predicted fractional increases in volume and Na⁺ excretions after ENaC inhibition.** Experimental and predicted fractional increases in volume and Na⁺ excretions in male and female rats at zeitgeber times 0 and 12 h.

## Discussion

Renal oxygen delivery and consumption, which determine kidney oxygen tension, are dependent on renal hemodynamics and metabolism, respectively. Active electrolyte transport is responsible for about 80% of the total renal oxygen consumption [41]. Therefore, renal tissue oxygenation fluctuates with alterations in Na⁺ reabsorption. The kidneys reabsorb nearly 99% of the filtered Na⁺. The proximal tubule reabsorbs approximately 60–70% of the filtered Na⁺ followed by the thick ascending loop of Henle which reabsorbs about 25–30%; the distal tubules reabsorb only 10% of the filtered Na⁺ [42]. Na⁺ reabsorption mediated by the sodium-potassium pump Na-K-ATPase on the basolateral membrane is the primary oxygen consuming process in the kidney. Na-K-ATPase transports 3 Na⁺ out of the cell in exchange for 2 K⁺ at the expense of 1 ATP molecule [43]. Hence, to reabsorb 99% of the filtered Na⁺, the kidneys require sufficient oxygen delivery to meet the large ATP demand.

The kidneys are well perfused but have low oxygen extraction. Thus, the kidneys perform high ATP-requiring transport activities in a low oxygen environment, particularly in the medulla where oxygen perfusion is relatively lower than the cortex, which makes this region especially prone to hypoxia. Oxygen consumption depends on GFR and transporter activities. Since these exhibit diurnal variations and sexual dimorphism, it is important to study the diurnal variations and sexual dimorphism in oxygen consumption to understand how the risk of renal hypoxia varies during the day and between the sexes.

In this study, we developed sex- and time-of-day specific computational models of rat kidney function to assess the diurnal variations in medullary oxygen consumption and oxygenation in male and female rats. The model predicted significant differences in oxygen

consumption during the day and between the sexes. Whole kidney total oxygen consumption was ~43% higher during the active period relative to the inactive period in both male and female rats, whereas that in the cortical and medullary segments were ~39% and ~48% higher during the active period relative to the inactive period. Oxygen consumption also varied between the sexes due to the sexual dimorphism of $Na^+$ transporter activities. Male rats showed higher oxygen consumption in the cortical segments (~1/3 higher), whereas female renal oxygen consumption was higher in the medullary segments (~1/5 higher).

Layton et al. developed a model for studying the sex differences in circadian control of kidney function in mice [27]. Rats and mice have species differences in tubular dimensions and transporter activities [44]. In addition, the circadian rhythms of different transporter activities also differ between the two rodent species [19]. For instance, NHE3 and ENaC activities in mice peak during the light (inactive) phase [22, 26], whereas for rats, these two transporters peak during the dark (active) phase [45]. Since, in rats, NHE3 and ENaC peak during the active period when the filtered load is also high, rat proximal tubules and distal tubules are able to reabsorb the high filtered $Na^+$ during the active phase. Because of this, the delivered $Na^+$ loads to the thick ascending limb and collecting duct during the active and inactive phases do not differ significantly (Fig 3). In contrast, since NHE3 and ENaC activities are near the nadir during the active phase in mice, we observe that the delivered $Na^+$ loads to the downstream segments differ significantly between the active and inactive phases (Fig 4 of [27]). Thus, differences in the circadian regulation of $Na^+$ transporter activities in rats and mice cause differences in their $Na^+$ reabsorption during the active and inactive phases.

Renal hypoxia plays an important role in the development of both acute kidney injury (AKI) and chronic kidney disease (CKD). Renal hypoxia results in AKI and CKD patients because of decreased renal oxygen supply and increased oxygen consumption. Recent studies have also reported interaction between circadian clocks and hypoxia [46–50]. Circadian clocks control diurnal oscillations in tissue oxygenation, which in turn synchronizes clocks through HIF1α [46, 47]. Clock components simultaneously modulate HIF1α levels and activity [48–50]. A recent study demonstrated that several core clock genes were phase-shifted in response to hypoxia in a tissue-specific manner, which caused inter-tissue circadian clock misalignment [51]. The present study utilized a simple balance equation to predict $p_{O2}$. Eq 4 assumes that oxygen supply and consumption are homogenous within the cortical and medullary compartments. However, anatomic studies have demonstrated that the nephrons and vessels in the medulla of rodent kidneys are organized in a highly structured manner [52–54]. The oxygen-carrying descending vasa recta in the inner stripe are isolated within tightly packed vascular bundles, separated from the metabolically demanding thick ascending limbs. Simulations using computational models that capture these 3D structures have yielded marked gradients in intrabundle and interbundle interstitial fluid oxygen tension [55–57]. Because of their high metabolic demand and their separation from the vascular bundles, the medullary thick ascending limbs operate near hypoxia [58]. The effect of the 3D architecture in the outer medulla on renal oxygenation as well as the heightened risk of hypoxia of the medullary TAL are not captured in the present model.

Loop diuretics are frequently used in the treatment of AKI and CKD. By inhibiting thick ascending limb active $Na^+$ transport, loop diuretics have been found to ameliorate medullary hypoxia [39]. Our model predicted that loop diuretics were significantly more effective in improving medullary oxygenation in female rats (by ~17.2% during the active period compared to the ~6.8% improvement in male rats). Also, the effect was higher when loop diuretic was administered during the active phase. These results highlight the importance of sex and time-of-day on physiological functions and drug administration [59]. These factors should be taken into account during biomedical research and modelling analysis. Drug absorption,

**Table 1. Peak times and oscillation amplitudes of model parameters regulated by circadian clocks.**

|  | Male amplitude (%) | Female amplitude (%) | Peak (ZT, 0 = 8 AM) (h) | References |
|---|---|---|---|---|
| GFR | 14 | 14 | 18 | [20] |
| NHE3 | 40 | 40 | 14 | [45] |
| SGLT1 | 20 | 20 | 14 | estimated |
| NKCC2 | 20 | 20 | 14 | estimated |
| NCC | 20 | 20 | 14 | [45] |
| ENaC | 56 | 56 | 14 | [45] |
| Renal blood flow | 20 | 20 | 10 | [38] |

metabolism, and excretion exhibit diurnal variation. This means that giving the same dose of a drug at different times of the day can result in different effects. Sex and time-specific models can be valuable tools for improving personalized medicine as they can be used to determine the optimal drug doses and the most suitable time for drug administration for each sex.

## Methods

We developed sex- and time-of-day specific models of the rat kidney function. A schematic diagram of the various cell types is given in Fig 1. The models are based on the epithelial cell-based model of solute transport in a rat kidney developed by Layton et al. [14, 35]. These models represent sexual dimorphism in tubular dimensions, single nephron glomerular filtration rate (SNGFR), and expression levels of apical and basolateral transporters in rats. In addition, the diurnal variation of GFR and transporter activities are modeled by varying them as sinusoidal functions of time. Model parameters that are regulated by circadian clocks are summarized in Table 1 and Fig 2.

### Model structure

The model for male or female rat kidney includes six types of nephrons: one superficial nephron and five juxtamedullary nephrons that are assumed to reach depths of 1, 2, 3, 4, and 5 mm into the inner medulla. We assume the superficial nephron to be 2/3 of the nephron population and the five juxtamedullary nephrons to be 0.4/3, 0.3/3, 0.15/3, 0.1/3, and 0.05/3 of the nephron population. The superficial nephron includes the proximal tubule, short descending limb, thick ascending limb, distal convoluted tubule, and connecting tubule segments. The juxtamedullary nephrons include all the segments of the superficial nephron with the addition of the long descending limbs and ascending thin limbs; these segments extend into the inner medulla and their lengths are determined by the type of juxtamedullary nephron being modeled. The connecting tubules of the five juxtamedullary nephrons and the superficial nephron coalesce into the cortical collecting duct. For male rats, the SNGFRs for the superficial and juxtamedullary nephrons are set to 30 and 45 nl/min, respectively, whereas the corresponding values for female rats are 24 and 36 nl/min, respectively.

Each nephron segment is represented as a tubule lined by a layer of epithelial cells. The segment and cell type (intercalated and principal cells) determine the type and abundance of transporters found on the apical and basolateral membranes of the cell. A paracellular pathway exists between neighboring cells. The models consider the following 15 solutes: $Na^+$, $K^+$, $Cl^-$, $HCO_3^-$, $H_2CO_3$, $CO_2$, $NH_3$, $NH_4^+$, $HPO_4^{2-}$, $H_2PO_4^-$, $H^+$, $HCO_2^-$, $H_2CO_2$, urea, and glucose. Male and female models differ in parameters describing membrane transporter and channel activities and paracellular permeabilities. The models are defined by a large system of coupled ordinary differential and algebraic equations for calculating transmembrane and paracellular

fluxes [60]. Osmotic and hydrostatic pressure differences are used to represent water fluxes. Transmembrane solute fluxes consist of passive and/or active components. An uncharged solute is passively transported across a membrane by a concentration gradient, whereas a charged solute is passively transported across an ion channel by an electrochemical potential gradient. In addition, two or more solutes may be actively and simultaneously transported across cotransporters, exchangers and/or ATP-driven pumps [22–25]. The model predicts the steady-state values of urine volume, urinary excretion rates of individual solutes, as well as luminal fluid flow throughout the nephron, hydrostatic pressure, membrane potential, luminal and cytosolic solute concentrations, and transcellular and paracellular fluxes through transporters and channels [14].

## Circadian rhythms in transport parameters

The peak times and oscillation amplitudes of NHE3, NCC, and ENaC are taken from a study conducted on male rats [45]. The peak times for NKCC2 and SGLT1 are assumed to be similar to NHE3, NCC, and ENaC due to lack of data on their diurnal variation in rats. The amplitudes of NKCC2 and SGLT1 are fitted such that the $Na^+$, $K^+$, and volume excretions match the experimental excretion values in [20] for male rats. Though female rats have lower volume, $Na^+$, and $K^+$ excretion rates relative to male rats, the rates are not significantly different [13]. Hence, we assume that the excretion rates of female rats are ~5% lower than those of male rats. We used the root mean squared error between the simulated and experimental excretion rates to fit the transporter activity amplitudes for NKCC2 and SGLT1. Since we did not find any studies on the circadian variation of these renal transporters in female rats, we assumed the same peak times and amplitudes between male and female rats. However, we have maintained the sex differences in the average transporter activities. For instance, the average NKCC2 activity in female rats is double that in male rats. The experimental and simulated volume, $Na^+$, and $K^+$ excretion rates for male and female rats are given in S3 Fig.

For nocturnal animals, ZT0 (lights phase) denotes the start of the rest phase and ZT12 (dark phase) denotes the start of the active phase. The diurnal variation in transporter activities (see Table 1) is represented as

$$X_p(t) = X_{p,0}\left(1 + \gamma_p \sin\left(\frac{2\pi(t + 6 - \theta_p)}{24}\right)\right),$$

where $t$ denotes the zeitgeber time (ZT), $X_{p,0}$ denotes the average activity, $\gamma_p$ denotes the fractional oscillation amplitude, and $\theta_p$ denotes the peak time. The circadian variation of GFR is represented similarly. Parameter values are specified in Table 1.

Model predictions at a given ZT (e.g., 14) were determined by setting the GFR and transporter activities at their ZT = 14 values, followed by the computation of the steady-state model solution. Since tubular flows have much shorter timescales compared with circadian oscillations, we have used a steady-state solution for our model.

## Oxygen consumption along the nephron

Oxygen consumption ($Q_{O2}$) consists of two parts: $Q_{O2}^{active}$, which represents the oxygen consumed to actively reabsorb $Na^+$ through the basolateral Na-K-ATPase pumps, and $Q_{O2}^{basal}$, which represents the oxygen consumed for other transport processes and intracellular biochemical reactions [35, 61]. Na-K-ATPase pumps out 3 moles of $Na^+$ using the energy from 1 mole of ATP; oxidative metabolism produces 5 moles of ATP per mole of $O_2$ consumed [62].

Thus, $Q_{O2}^{active}$ is determined as

$$Q_{O2}^{active} = T_{Na}^{active}/15, \tag{1}$$

where $T_{Na}^{active}$ is the rate of Na$^+$ transport across Na-K-ATPase pumps [35, 61].

The whole kidney basal-to-total $Q_{O2}$ ratio is about 25–30% in rats [63]. We assumed that $Q_{O2}^{basal}$ is fixed and equal to 25% of the total $Q_{O2}$ under baseline conditions [35, 61], such that,

$$Q_{O2}^{basal} = 0.25(Q_{O2}^{basal} + Q_{O2}^{active*}) = (0.25/0.75)Q_{O2}^{active*}, \tag{2}$$

where * denotes baseline conditions.

The efficiency of oxygen utilization is defined as the number of moles of Na$^+$ reabsorbed per mole of O$_2$ consumed [35, 61]:

$$T_{Na}^{total}/Q_{O2}^{total} = \frac{T_{Na}^{active} + T_{Na}^{passive}}{Q_{O2}^{active} + Q_{O2}^{basal}}, \tag{3}$$

where $T_{Na}^{passive}$ denotes the rate of passive Na$^+$ reabsorption.

## Estimation of partial pressure of oxygen ($p_{O2}$)

The renal outer medullary partial pressure of oxygen ($p_{O2}$, mmHg) is estimated using the following equation:

$$p_{O2} = \frac{1}{\alpha}(D_{O2} - X_{O2} - Q_{O2}), \tag{4}$$

where $D_{O2}$ denotes oxygen delivery (µmol min$^{-1}$) from the descending vasa recta, $X_{O2}$ denotes oxygen shunting (µmol min$^{-1}$) between the descending and ascending vasa recta, $Q_{O2}$ denotes the oxygen consumption in the renal outer medulla (µmol min$^{-1}$), and α represents the pressure to µmol conversion factor (µmol min$^{-1}$ mmHg$^{-1}$).

Oxygen delivery, $D_{O2}$, is calculated as [64]

$$D_{O2} = MRBF \times CaO_2, \tag{5}$$

where $MRBF$ denotes medullary renal blood flow (mL min$^{-1}$) and $CaO_2$ denotes the arterial oxygen content. We assumed the mean $MRBF$ for male rats to be 2.26 mL min$^{-1}$ (assuming a mean kidney weight of 1.189 g) [65, 66]. In the model, the single nephron GFR for male rats is assumed to be 25% higher than that for female rats. Hence, we assumed the mean $MRBF$ for female rats to be 1.81 mL min$^{-1}$. The arterial oxygen content is calculated as [64]

$$CaO_2 = O_2\ bound\ to\ Hb + O_2\ dissolved\ in\ plasma,$$

$$CaO_2 = (Hb \times 1.34 \times SaO_2) + (0.003 \times PaO_2), \tag{6}$$

where $Hb$ denotes the amount of hemoglobin which is 146 and 141 g L$^{-1}$ in male and female rats, respectively [67]. Amount of oxygen carried by 1 g of Hb is 1.34 mL [64]. $SaO_2$ denotes the arterial oxygen saturation which we assume as 95%. Oxygen dissolved in plasma is 0.003 mL O$_2$ per 100 mL blood per mmHg [64]. $PaO_2$ denotes the arterial partial pressure of oxygen which is ~88 mmHg in rats [68].

Approximately 2.6% of the total oxygen delivered through the descending vasa recta is shunted to the ascending vasa recta [69]. Hence,

$$X_{O2} = 0.026*D_{O2}. \tag{7}$$

Mean $p_{O2}$ in outer medulla is 15–30 mmHg [6]; we assumed this as 22.5 mmHg. We used the peak time and oscillation amplitude of the outer medullary $p_{O2}$ as 13 h (ZT) and 8%, respectively [37]. We assumed the same medullary $p_{O2}$ for male and female rats [32].

## Simulating loop diuretics

Inhibition of active transport along the loop of Henle with loop diuretics, such as furosemide, can lower oxygen consumption and increase medullary oxygen tension [39]. Loop diuretics inhibit the Na$^+$ transporter, NKCC2, which is expressed on the apical membrane of the thick ascending limbs. We lowered the interstitial fluid concentrations of selected solutes [70] based on the assumption that the loop diuretic has been administrated for a considerable duration to significantly decrease the generation of axial osmolality gradient [29, 70]. In addition, since targeted deletion of NKCC2 significantly attenuates the tubuloglomerular feedback response [71], we assumed that the SNGFR remained at baseline values, consistent with an experimental study in rats [72]. Furosemide, a loop diuretic, is mainly secreted into the proximal tubule by the organic anion transporter-1 (OAT1) [73]. OAT mRNA levels in the kidney exhibit circadian rhythms, peaking in the late light phase and early dark phase [74], which results in time-dependent excretion of OAT substrates such as furosemide. A reduction in renal excretion of furosemide was observed on deleting the clock gene, Bmal1, in mice [74]. To represent the effect of the circadian variation in renal OAT expression, we assumed that 80% of NKCC2 activity is inhibited in the dark (active) phase and 70% of NKCC2 activity is inhibited in the light (inactive) phase.

## Supporting information

**S1 Fig. Time profiles of filtered values.** Time profiles of (A) glomerular filtration rate (GFR), (B) filtered sodium load, and (C) filtered potassium load.
(TIF)

**S2 Fig. Predicted segmental and regional efficiency of oxygen utilization.** Predicted oxygen utilization efficiency in (A, B, C) the proximal tubules (PT), thick ascending limbs (TAL), and distal tubules (DT); and (D, E, F) the cortical segments, medullary segments, and whole kidney of male and female rats at zeitgeber times 2, 6, 14 and 18 h. The values are given per kidney.
(TIF)

**S3 Fig. Experimental and simulated excretion rates.** Experimental and simulated volume (A, B), Na$^+$ (C, D), and K$^+$ (E, F) excretion rates in male and female rats at zeitgeber times 2, 6, 14, and 18 h. Circadian oscillations in selected transporter activities were fitted so that the predicted excretion rates are in sufficient agreement with the experimental values.
(TIF)

## Author Contributions

**Conceptualization:** Anita T. Layton.

**Data curation:** Pritha Dutta, Anita T. Layton.

**Formal analysis:** Pritha Dutta, Anita T. Layton.

**Funding acquisition:** Anita T. Layton.

**Methodology:** Pritha Dutta, Anita T. Layton.

**Project administration:** Anita T. Layton.

**Resources:** Anita T. Layton.

**Software:** Pritha Dutta, Anita T. Layton.

**Supervision:** Anita T. Layton.

**Validation:** Pritha Dutta.

**Visualization:** Pritha Dutta.

**Writing – original draft:** Pritha Dutta.

**Writing – review & editing:** Pritha Dutta, Anita T. Layton.

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
