## [Decision Letter · Decision Letter 0]

24 Aug 2023

PONE-D-23-10988Sex and circadian clock: emerging determinants of renal metabolic demandPLOS ONE

Dear Dr. Dutta,

Thank you for submitting your manuscript to PLOS ONE. After careful consideration, we feel that it has merit but does not fully meet PLOS ONE’s publication criteria as it currently stands. Therefore, we invite you to submit a revised version of the manuscript that addresses the points raised during the review process.

One of the reviewers has noted a similarity between your current manuscript and a previously published work by Layton (doi.org/10.1152/ajprenal.00227.2022). It is crucial to differentiate the two articles clearly and highlight the advancements and novel contributions your current work brings. In your revised manuscript, please consider including a dedicated section that provides a comprehensive comparison between the two articles, emphasizing the distinct elements of your present study.

We look forward to receiving your revised manuscript.

Kind regards,

Toryn Poolman

Academic Editor

PLOS ONE

Journal Requirements:

"This study is supported in part by grants from the Natural Sciences and Engineering Research Council (NSERC) and Canadian Institutes of Health Research (CIHR) of Canada to A.T. Layton."

Reviewers' comments:

Reviewer's Responses to Questions

**Comments to the Author**

1. Is the manuscript technically sound, and do the data support the conclusions?

Reviewer #1: Yes

Reviewer #2: Partly

2. Has the statistical analysis been performed appropriately and rigorously? 

Reviewer #1: Yes

Reviewer #2: N/A

3. Have the authors made all data underlying the findings in their manuscript fully available?

Reviewer #1: Yes

Reviewer #2: Yes

4. Is the manuscript presented in an intelligible fashion and written in standard English?

Reviewer #1: Yes

Reviewer #2: Yes

5. Review Comments to the Author

Reviewer #1: This is a well-written study by Dutta and Layton investigating sex and diurnal variation in renal oxygen consumption using model simulation.

The title should be more specific to study findings including a highlight that the results are driven from model simulation.

How does the currently-used model simulation incorporate published evidence for sex differences in the length of different nephron segments which can account for balancing some of the evident sex differences in transporters abundance. In particular, Tahaei et al. showed that female rodents express a greater density of NCC transporters in a shorter DCT structure to protect Na+ balance (PMID: 32924546, AJP Renal Physiol, 2020).

Figure 2D displays no differences between sexes in ENaC activity throughout different times of the day. This finding contradicts with reported differences by Soliman and colleagues (AJP Regu, 2020, PMID 31913682) showing that benzamil natriuretic response is greater in male compared with female rats whether given at the beginning of the inactive period or active period. However, no difference in benzamil response is present between ZT0 and ZT12 dosing in either sex. Discuss this discrepancy and how these physiological data can be incorporated into the current model to improve the data prediction.

The expression of estrogen and androgen receptors has been shown in different nephron segments. Discuss the potential of incorporating sex hormone receptor expression into the model especially as estrogen receptors has been shown to impact Na transporters including NKA and NCC (Gohar et al., JAHA, 2020 “PMID: 32390531” and Cheng et al, JASN, 2019 “PMID: 31253651”)

Fig 7 data would be better incorporated in a tabulated format.

Fig. 3: use axis cutting to allow displaying different parts of the scale, rather than figure duplication.

Fig. 4-8 are displaying circadian differences in each sex separately. For the pupose of the current study, consider displaying male-female timepoints on the same figure to allow both sex and time of the data comparisons.

Reviewer #2: Dutta and Layton propose in this article a model supposed to predict the sex-specific diurnal kidney activity and metabolism. Build on existing model for sex and time difference, this model also incorporates oxygen consumption. This is one of the first issue with this article: many parts of the article (including figures and results) appear very similar to a previous article published by Layton (doi.org/10.1152/ajprenal.00227.2022). The exact difference (and similarities) between the two articles needs to be clearly identified and explained.

One other issue is the many assumptions made by the author (e.g., sex difference in GFR). It would be great to provide some rational about these assumptions. In the same idea, many parameters in the model are “estimated” but there is no explanation or rational for this estimation.

Figure 7 shows the experimental and predicted pO2. What is the origin of the experimental value that appear anyway identical in males and females? There is no description in the method.

While the idea of the simulation of the action of diuretic is interesting, it would be great to validate the estimation by experimental results.

6. PLOS authors have the option to publish the peer review history of their article (what does this mean?). If published, this will include your full peer review and any attached files.

Reviewer #1: No

Reviewer #2: No

---

## [Author Response · Author response to Decision Letter 0]

20 Sep 2023

We thank the reviewers for their helpful comments, which have led to significant improvements to the manuscript. 

Summary of major changes:

1. In the original submission, the circadian profiles of the transporter activities were fitted, separately for the male and female rats, to their respective excretion data. In the revision, we improved the parameter specification by matching the peak times and amplitudes of the transporter profiles of the male rat kidney to data in Ref. [46]. That gives those model parameters a much stronger, data-driven basis. Unfortunately, analogous circadian data isn’t available for the female rats. Nonetheless, assuming the same circadian oscillations to both sexes, the models predict urinary excretions that are consistent with data in both sexes. Thus, in the revision, the same circadian oscillations in transporter activities are assumed for male and female (Fig. 2). 

2. Because the transporter oscillations peak at ZT14 and GFR at ZT18, we now plot results at those times, together with the respective nadirs.

Reviewer #1: 

This is a well-written study by Dutta and Layton investigating sex and diurnal variation in renal oxygen consumption using model simulation.

1. The title should be more specific to study findings including a highlight that the results are driven from model simulation.

Ans: We thank the reviewer for the suggestion. We have modified the title in the revised manuscript as “Sex and circadian regulation of metabolic demands in the rat kidney: a modeling analysis”.

2. How does the currently used model simulation incorporate published evidence for sex differences in the length of different nephron segments which can account for balancing some of the evident sex differences in transporters abundance. In particular, Tahaei et al. showed that female rodents express a greater density of NCC transporters in a shorter DCT structure to protect Na+ balance (PMID: 32924546, AJP Renal Physiol, 2020).

Ans. The model accounts for sex differences in the length and diameter of different nephron segments. For details, please refer to (doi.org/10.1152/ajprenal.00352.2019). The following table gives the female to male ratios of the lengths of the main nephron segments.

Nephron segment Female/Male ratio of length

Proximal tubule 0.8

Thick ascending limb 0.9

Distal convoluted tubule 0.9

Connecting tubule 0.85

Collecting duct 0.85

3. Figure 2D displays no differences between sexes in ENaC activity throughout different times of the day. This finding contradicts with reported differences by Soliman and colleagues (AJP Regu, 2020, PMID 31913682) showing that benzamil natriuretic response is greater in male compared with female rats whether given at the beginning of the inactive period or active period. However, no difference in benzamil response is present between ZT0 and ZT12 dosing in either sex. Discuss this discrepancy and how these physiological data can be incorporated into the current model to improve the data prediction.

Ans: Thank you for bringing to our notice this interesting experimental data. Soliman et al. reported that ENaC inhibition caused greater diuretic and natriuretic responses in male rats compared to female rats at the beginning of both active and inactive phases (doi.org/10.1152/ajpregu.00060.2019). We simulated the effect of ENaC inhibition by inhibiting ENaC activity in our model by 100%. Figure 9 compares the predicted fractional increases in volume and Na+ excretions after ENaC inhibition at the beginning of the inactive (ZT = 0 h) and active (ZT = 12 h) phases in male and female rats with the corresponding fractional increases reported by Soliman et al. (doi.org/10.1152/ajpregu.00060.2019). ENaC inhibition causes higher fractional increases in volume and Na+ excretions in male rats compared to female rats during the beginning of both inactive and active phases. This is because males have higher delivered volume and Na+ loads to the connecting tubule, the segment from which ENaC is the main Na+ transporter. The Na+ loads delivered to the male connecting tubule are 98% and 61% higher during zeitgeber times 0 and 12 h, respectively, compared to that delivered to the female connecting tubule. The corresponding volume loads delivered to the male connecting tubule are 85% and 57% higher, respectively. For this reason, inhibiting ENaC causes greater diuretic and natriuretic responses in male rats compared to females. Another interesting observation is that the diuretic and natriuretic responses are significantly higher when ENaC is inhibited at the beginning of inactive phase compared to the beginning of the active phase in male rats. This is because the Na+ and volume loads delivered to the connecting tubule are 35% and 28% higher, respectively, during ZT = 0 h compared to ZT = 12 h. By contrast, for females, the corresponding loads are only 9% and 4.6% higher, respectively. This is because the GFR is the same at ZT = 0 h and ZT = 12 h (Fig 2). However, the Na+ transporters (NHE3, NKCC2, NCC) are near nadir at 0 h (Fig 2). Hence, less Na+ and water gets reabsorbed along the early nephron segments at 0 h. Since, males have 25% higher GFR than females, the delivered load to the connecting tubule is significantly higher in male rats.

Fig 9. Comparison of experimental and predicted fractional increases in volume and Na+ excretions after ENaC inhibition. Experimental and predicted fractional increases in volume and Na+ excretions in male and female rats at zeitgeber times 0 and 12 h.

4. The expression of estrogen and androgen receptors has been shown in different nephron segments. Discuss the potential of incorporating sex hormone receptor expression into the model especially as estrogen receptors has been shown to impact Na transporters including NKA and NCC (Gohar et al., JAHA, 2020 “PMID: 32390531” and Cheng et al, JASN, 2019 “PMID: 31253651”)

Ans. Our models take into account the effect of sex hormones as the female rat model has higher NCC, Na-K-ATPase, and ENaC activities than the male rat model. For details, please refer to (doi.org/10.1152/ajprenal.00352.2019). The following table gives the female to male ratios of transporter activities along different nephron segments.

Transporter (nephron segment) Female/Male ratio

NCC (DCT) 2

Na-K-ATPase (TAL) 2

Na-K-ATPase (DCT, CNT) 2

Na-K-ATPase (CCD) 1.5

Na-K-ATPase (OMCD) 1.1

Na-K-ATPase (IMCD) 1.2

ENaC (DCT) 2

ENaC (CNT) 1.3

ENaC (CCD) 1.5

ENaC (OMCD) 1.2

5. Fig 7 data would be better incorporated in a tabulated format.

Ans: We have updated Fig 7 by displaying the male and female pO2 values in the same figure for easier comparison.

6. Fig. 3: use axis cutting to allow displaying different parts of the scale, rather than figure duplication.

Ans: We have updated Fig 3 according to the reviewer’s suggestion.

7. Fig. 4-8 are displaying circadian differences in each sex separately. For the purpose of the current study, consider displaying male-female timepoints on the same figure to allow both sex and time of the data comparisons.

Ans: We have updated Fig 4-8 according to the reviewer’s suggestion.

Reviewer #2: 

Dutta and Layton propose in this article a model supposed to predict the sex-specific diurnal kidney activity and metabolism. 

1. Build on existing model for sex and time difference, this model also incorporates oxygen consumption. This is one of the first issue with this article: many parts of the article (including figures and results) appear very similar to a previous article published by Layton (doi.org/10.1152/ajprenal.00227.2022). The exact difference (and similarities) between the two articles needs to be clearly identified and explained.

Ans: We thank the reviewer for this suggestion. We have included the differences between the rat and mice models in the Discussion section of the revised manuscript.

The article (doi.org/10.1152/ajprenal.00227.2022) reports the sex differences in circadian control of kidney function in mice. In our model, we studied the sex differences in circadian control of kidney function and renal oxygen consumption in rats. Rats and mice have species differences in tubular dimensions and transporter activities (doi.org/10.3389/fphys.2022.991705). In addition, the circadian rhythms of different transporter activities also differ between the two rodent species. For instance, NHE3 and ENaC activities in mice peak during the light (inactive) phase (doi.org/10.1152/ajprenal.00014.2020, doi.org/10.1111/j.1523-1755.2005.00218.x), whereas for rats, these two transporters peak during the dark (active) phase (doi.org/10.3881/j.issn.1000-503X.2015.06.010, doi.org/10.1016/j.freeradbiomed.2018.01.018). Since, in rats, NHE3 and ENaC peak during the active period when the filtered load is also high, rat proximal tubules and distal tubules are able to reabsorb the high filtered Na+ during the active phase. Because of this, the delivered Na+ loads to the thick ascending limb and collecting duct during the active and inactive phases do not differ significantly (Fig 3 of our manuscript). In contrast, since NHE3 and ENaC activities are near the nadir during the active phase in mice, we observe that the delivered Na+ loads to the downstream segments significantly differ between the active and inactive phases (Fig 4 of doi.org/10.1152/ajprenal.00227.2022). Thus, differences in the circadian regulation in Na+ transporter activities in rats and mice cause differences in their Na+ reabsorption during the active and inactive phases.

2. One other issue is the many assumptions made by the author (e.g., sex difference in GFR). It would be great to provide some rational about these assumptions.

Ans: Sex differences in GFR and transporter activities are based on experimental data. Male rats have been found to have higher GFR than female rats (doi.org/10.1152/ajprenal.1988.254.2.F223, doi.org/10.1038/ki.1988.206). Veiras et al. reported sex differences in the expression and abundance of different renal transporters (doi.org/10.1681/ASN.2017030295).

3. In the same idea, many parameters in the model are “estimated” but there is no explanation or rational for this estimation.

Ans: We have updated the model parameters in the revised manuscript (Table 1). The peak times and oscillation amplitudes of NHE3, NCC, and ENaC are taken from a study conducted by Zhang et al. on male rats (doi.org/10.3881/j.issn.1000-503X.2015.06.010). The peak times for NKCC2 and SGLT1 are assumed to be similar to NHE3, NCC, and ENaC due to lack of data on their diurnal variation in rats. The amplitudes of NKCC2 and SGLT1 are fitted such that the Na+, K+, and volume excretions match the experimental excretion values in (doi.org/10.3109/07420529409057246). Since we did not find any studies on the circadian variation of these renal transporters in female rats, we assumed the same peak times and amplitudes between male and female rats. However, we have maintained the sex differences in the average transporter activities. For example, the average NKCC2 activity in female rats is double that in male rats. 

4. Figure 7 shows the experimental and predicted pO2. What is the origin of the experimental value that appear anyway identical in males and females? There is no description in the method.

Ans: The experimental pO2 values are generated using the oscillation amplitude and peak time reported in (doi.org/10.3389/fphys.2017.00205) for male rats. Since a similar renal oxygen tension has been observed in age-matched male and female rats (doi.org/10.18383/j.tom.2020.00022), we assumed the same pO2 values for male and female rats. The calculation of pO2 is explained in Methods: Estimation of partial pressure of oxygen (\\mathbf{p}_{\\mathbf{O2}}).

5. While the idea of the simulation of the action of diuretic is interesting, it would be great to validate the estimation by experimental results.

Ans: We understand the reviewer’s concern about validating our simulation results with experimental data. However, we did not find any sex and time-of-day specific study on the effect of loop diuretics on renal oxygenation. However, based on Reviewer 1’s suggestion, we have included in the revised manuscript simulation results and explanations for the observed sex differences in the diuretic and natriuretic responses on ENaC inhibition during the beginning of the active and inactive phases reported by Soliman et al. (doi.org/10.1152/ajpregu.00060.2019).

---

## [Decision Letter · Decision Letter 1]

12 Oct 2023

Sex and circadian regulation of metabolic demands in the rat kidney: a modeling analysis

PONE-D-23-10988R1

Dear Dr. Dutta,

We’re pleased to inform you that your manuscript has been judged scientifically suitable for publication and will be formally accepted for publication once it meets all outstanding technical requirements.

Kind regards,

Toryn Poolman

Academic Editor

PLOS ONE

Additional Editor Comments (optional):

Reviewers' comments:

Reviewer's Responses to Questions

**Comments to the Author**

1. If the authors have adequately addressed your comments raised in a previous round of review and you feel that this manuscript is now acceptable for publication, you may indicate that here to bypass the “Comments to the Author” section, enter your conflict of interest statement in the “Confidential to Editor” section, and submit your "Accept" recommendation.

Reviewer #1: All comments have been addressed

Reviewer #2: All comments have been addressed

2. Is the manuscript technically sound, and do the data support the conclusions?

Reviewer #1: Yes

Reviewer #2: Yes

3. Has the statistical analysis been performed appropriately and rigorously? 

Reviewer #1: Yes

Reviewer #2: Yes

4. Have the authors made all data underlying the findings in their manuscript fully available?

Reviewer #1: Yes

Reviewer #2: Yes

5. Is the manuscript presented in an intelligible fashion and written in standard English?

Reviewer #1: Yes

Reviewer #2: Yes

6. Review Comments to the Author

Reviewer #1: The authors have adequately addressed my comments, however incorporating information about sex hormone receptors along the nephron would be a good addition.

Reviewer #2: Authors have adequatly answered all the querries made by the reviewers. The article is now acceptable for publication.

7. PLOS authors have the option to publish the peer review history of their article (what does this mean?). If published, this will include your full peer review and any attached files.

Reviewer #1: No

Reviewer #2: No

---

## [Editor Report · Acceptance letter]

12 Jun 2024

PONE-D-23-10988R1 

PLOS ONE

Dear Dr. Dutta, 

I'm pleased to inform you that your manuscript has been deemed suitable for publication in PLOS ONE. Congratulations! Your manuscript is now being handed over to our production team.

Kind regards, 

on behalf of

Dr. Toryn Poolman 

Academic Editor

PLOS ONE